# Membrane fission via transmembrane contact

Russell K. W. Spencer [1] ✉, Isaac Santos-Pérez[2], Izaro Rodríguez-Renovales[3,4], Juan Manuel Martinez Galvez[4,5], Anna V. Shnyrova [4,5] ✉ & Marcus Müller [1] ✉

Division of intracellular organelles often correlates with additional membrane wrapping, e.g., by the endoplasmic reticulum or the outer mitochondrial membrane. Such wrapping plays a vital role in proteome and lipidome organization. However, how an extra membrane impacts the mechanics of the division has not been investigated. Here we combine fluorescence and cryo-electron microscopy experiments with self-consistent field theory to explore the stress-induced instabilities imposed by membrane wrapping in a simple double-membrane tubular system. We find that, at physiologically relevant conditions, the outer membrane facilitates an alternative pathway for the inner-tube fission through the formation of a transient contact (hemi-fusion) between both membranes. A detailed molecular theory of the fission pathways in the double membrane system reveals the topological complexity of the process, resulting both in leaky and leakless intermediates, with energies and topologies predicting physiological events.

Transient contacts between the membranes of different organelles have emerged as a new paradigm in intracellular trafficking[1–3]. Some of such contacts have been postulated to have an active role in organelles' division[4,5]. This is the case for endoplasmic-reticulum (ER) contacts that result in partial wrapping around organelles, such as the mitochondrion or the endosome[4,6,7]. Such contacts help organize proteins and lipids at the fission site, yet the additional membrane layer may also have a mechanical role in membrane constriction and fission. While it seems counterproductive to constrict two membranes instead of one, the formation of a multilayered structure may result in several advantages for membrane remodeling during the scission process. First, the extra membrane makes the membrane system thicker, potentially increasing the surface for the assembly of specialized protein machineries. Second, the stress (e.g., tension or pulling forces) can be applied via the outer membrane, making the extra membrane layer act as a mechanical transducer. The highly dynamic ER network, driven by motors and active filaments, might be a generic source of such stress on the outer membrane[8]. A similar rationale applies to the mitochondrial double membrane. The apparatus driving

the mitochondrial fission localizes on the outer mitochondrial membrane[9], while the inner mitochondrial membrane feels and reacts to the stresses transduced through the outer one. However, how stress transduction works in double-membrane (DM) systems under constriction has remained completely unexplored.

There is a consensus on the fission mechanism of single-membrane (SM) tubes[10–15]: Initially, a local constriction squeezes out the inner monolayer, and the outer monolayer collapses into a metastable, worm-like micelle (WLM). The thermally activated rupture of this WLM completes fission[10–15]. Therefore, our mechanistic understanding of SM fission establishes the lipid monolayers' elastic properties, stress and stress-driven instabilities as key components of the process[11–13,16–19] This paradigm emerges from extensive studies of lipid or minimal proteo-lipid systems, both experimental and theoretical. In contrast, the fundamental basis for the case of DM fission is virtually absent. However, such basic knowledge is a requirement to approach more complex processes of biological DM fission.

Here, we developed a toolbox to analyze lipid membrane remodeling pathways during DM fission at its fundamental level.

[1]Institute for Theoretical Physics, Georg-August University, Göttingen, Germany. [2]Electron Microscopy and Crystallography Platform, Center for Cooperative Research in Biosciences (CIC bioGUNE), Derio, Spain. [3]BREM Basque Resource for Electron Microscopy, Leioa, Spain. [4]Instituto Biofisika (CSIC, UPV/EHU), Barrio Sarriena, Leioa, Spain. [5]Department of Biochemistry and Molecular Biology, University of the Basque Country, Leioa, Spain. ✉e-mail: russell.spencer@uni-goettingen.de; anna.shnyrova@ehu.eus; mmueller@theorie.physik.uni-goettingen.de

We simplified the double-membrane system into a double-membrane (DM) tube. We then used osmotic stress or an increase in membrane tension (a well-known co-factor in single-membrane (SM) fission[14,20–23]) as membrane constriction factors, mimicking constriction brought about by specialized proteins during DM fission. The changes in membrane connectivity and topology observed by a combination of fluorescence and cryo-electron (cryoEM) microscopies are further investigated by self-consistent field theory (SCFT), providing detailed membrane remodeling pathways. Taken together, our approaches provide a description of DM fission mechanisms and allow for further estimates of membrane remodeling pathways driven by different specialized protein machineries in the cell.

Molecular theories[24,25] (such as SCFT) have proven successful in assessing lipid membrane remodeling events, such as lipid membrane fusion and fission[16,26–31], formation of the stalk or hemifusion (HF) diaphragm[27–31] and pores[28,29,31,32]. This approach allows studying membranes as emergent properties of lipid statistics without needing to expend computational effort tracking individual molecules, as would be required in particle simulations. Compared with molecular-dynamics simulation[18,19,26,33–40], SCFT is computationally faster, easier to apply to a broad parameter space and gives direct access to free energies while still including molecular degrees of freedom, which are lacking in, for example, Helfrich models[17,27,41–46]. Combined with the string method, which defines an optimal reaction coordinate in terms of local concentration changes [37,38], it results in a potent tool for examining topological changes and their free-energy profiles, including the large-scale rearrangements involved in membrane fission. This theoretical method has been successfully applied to small-scale membrane transformations, e.g., pore and stalk formation[26,27,32]. Here we used it for the first time to explore the transformations of the scale of the multilayered systems.

We start by analyzing the fission of SM and DM tubes, finding that a second membrane makes fission happen more readily. Through theoretical modeling, we find that the presence of a second membrane enables another, more favorable pathway for the fission of the inner membrane tube, which implies a transient intermembrane contact. This finding has major implications for our understanding of the remodeling of multilayered lipid membrane systems and provides the basis for predictions of the membrane dynamics in multilayer cellular systems, from transmembrane contacts to mitochondrial division.

## Results

### The outer membrane facilitates inner membrane fission in DM systems

We first observed SM and DM tube scission using real-time fluorescence microscopy. The tubes were prepared by "rolling" a multilayered lipid membrane reservoir on top of a SU8 polymer micropillar array in a microfluidic chamber[47] (see "Methods", Supplementary Fig. 1a). The limited amount of membrane deposited on the pillars set the tubes' tension, σ, which, in turn, determines the tube's radius, $r_{NT}$, as $r_{NT} = \sqrt{\frac{\kappa}{2\sigma}}$, where κ is the bending rigidity of the membrane dictated by its composition[48]. As the tube's membrane contains a fixed amount of a fluorophore, the cylindrical geometry of the tube could be quantified upon calibration of membrane area/fluorescence signal[47,48] of a single supported bilayer with the same composition and fluorophore content. The DM tubes form stochastically in this system, and their formation can be favored by supplying an extra membrane reservoir to the pillars[47]. The lamellarity of the tubes can be established by analyzing the distribution of integral fluorescence per unit length of the NTs[47] (see "Methods", Supplementary Fig. 1b). The narrow distribution of tensions (and thus tube radii) leads to two well-defined peaks in such a distribution corresponding to SM and DM tubes, respectively (Supplementary Fig. 1b). The maximum value of the DM peak doubles that of the SMs, indicating membranes proximity in the DM systems. CryoEM observations (Fig. 2b, Supplementary Fig. 2) further confirmed a narrow intermembrane distance for DM tubes with radii below 50 nm. Hence, in moderately curved DM tubes, the membranes are in close apposition, opening the possibility for intermembrane contacts (IMCs) upon further constriction.

Once the tubes were formed, we perfused the system with a hyperosmotic solution that caused the tubes to constrict to a critical radius, followed by their fission (Fig. 1). While for SM tubes this process has one intermediate (Fig. 1a), in DM tubes, the fission of the two tubes occurred sequentially, with the inner tube presumably undergoing fission first (Fig. 1b). We compared the radii of the SM and DM tubes before fission, taking a constant intermembrane space of 2 or 5 nm between the membranes of the DM tube (in agreement with cryoEM observations for constricted DM tubes, see Fig. 2 and Supplementary Fig. 2). Such analysis indicated that the inner tube of the DM system requires less constriction (pre-fission radius, $r_{DM} = 6.7 \pm 0.4$ nm (mean ± SD, $n = 3$ independent experiments)) to undergo fission as compared to a SM tube (pre-fission radius, $r_{SM} = 5.1 \pm 0.2$ nm (mean ± SD, $n = 3$

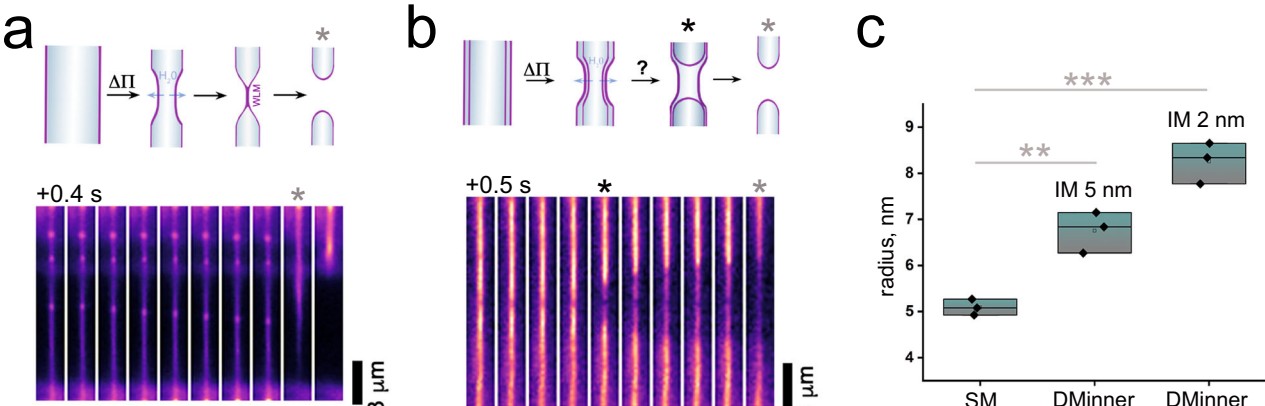

**Fig. 1 | Osmotically driven scission of DM and SM tubes.** Time sequence of SM (**a**) and DM tube scission (**b**) upon constriction by osmotic stress. The membrane is shown in pseudocolor. Cartoons illustrate the steps in the scission process in each case, while stars mark scission events. See also Supplementary Movie 1.
**c** Quantification of the radii of the SM and the inner membrane of the DM tubes right before scission ($n = 3$ independent experiments; each point represents a

tube). Inner DM radii were estimated assuming a constant intermembrane (IM) distance between opposing hydrophobic heads region as 2 or 5 nm, respectively (see "Methods"). ** statistically different at the 0.01 level; *** statistically different at the 0.001 level (unpaired two-sided two sample t-test, equal variance assumed; Box plots indicate median (middle line), 25th and 75th percentile (box), and outliers (whiskers)). Source data are provided as a Source data file.

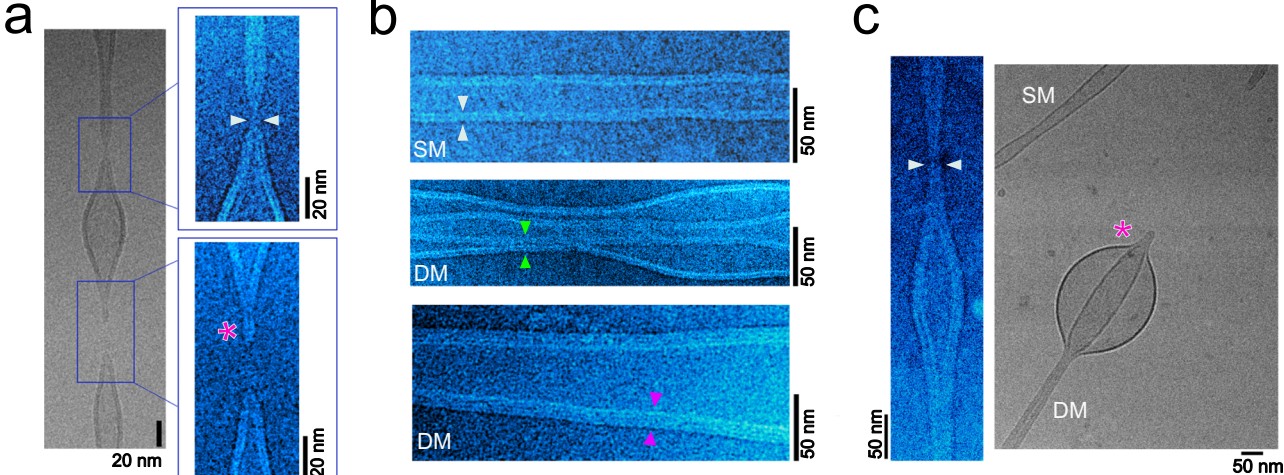

**Fig. 2 | CryoEM snapshots of tubes constricted by hydrodynamic flow.**
**a** Representative CryoEM micrograph of a SM tube under constriction undergoing fission. The insets show the formation of WLMs immediately before (upper image, white arrowheads) and after membrane scission occurs (lower image, *). Pseudocolor (cyan hot) is used for clarity. See Supplementary Fig. 1d. Scission, WLM, and HF intermediates were detected in 16, 25, and 20% of SM tubes, respectively (94 SM tubes, 7 independent experiments). **b** CryoEM snapshot of SM and DM tubes under constriction. White arrowheads indicate SM thickness (5.1 ± 0.5 nm (mean ± SD),

$n = 111$ random measurements, 18 tubes, 3 independent experiments). Green and magenta arrowheads point to proximal membranes of DM tubes under constriction. Pseudocolor is used for clarity. **c** IMC and fission in DM tubes. Left: representative image of a DM tube with an IMC (white arrowheads), i.e., with intermembrane distance below 0.5 nm. IMCs were detected in 9 out of 16 DM tubes (7 independent experiments). Right: DM tube fission. Simultaneous retraction of both membranes of a DM tube upon fission is indicated by (*). Fission was detected in 4 out of 16 DM tubes. Analysis is provided as a Source data file.

independent experiments)). Therefore, the presence of the outer tube facilitates the scission of the inner tube in the DM system.

## The low energy pathway for DM fission confirms leverage by the outer membrane

To rationalize the experimental findings, we developed a corresponding theoretical setup consisting of a SM or a DM tubular system. We used SCFT to find stable and metastable membrane configurations at a fixed membrane tension, such as SM and DM tubes, and metastable intermediate topologies. The tension is controlled by setting the lipid chemical potential, simulating a connection to a lipid reservoir in the experimental system. In turn, tension controls the tube's geometry, as the tube's radius is inversely proportional to the square root of tension[49]. The reservoir is the same for the inner and the outer tube, matching the in vitro DM tube production method described above. SCFT allows us to include molecular details by simulating the configurational statistics of lipids and their self-assembly while simultaneously computing the free energy. Further, in conjunction with the string method (see "Methods" and Supplementary Information), it enables us to construct the minimum free-energy path (MFEP) connecting metastable states, i.e., local minima of the free energy. The MFEP gives us intermediate configurations (transformation mechanism) and the free-energy changes as the system transforms and, thus, the barriers along each pathway.

Our calculations indicate that at relatively high tension (1 Dyn/cm), that corresponds to a physiologically relevant constriction of the tube (e.g., brought by specialized protein machinery), both SM tubes and the inner tubes of DM systems may undergo fission through the canonical fission pathway, where after initial high, local constriction, fission proceeds into two capped tubes connected by WLM or stalk, formed by the outer leaflet of the membrane tube. Subsequently, this WLM ruptures, completing the fission reaction (Fig. 3a). However, the initial DM tube's radius set by 1 Dyn/cm tension is relatively large (8 nm inner tube radius), as are the free-energy barriers along the canonical fission pathway. Thus, more local constriction is required to squeeze the tube to the critical constriction, beyond which the WLM forms. We found no significant decrease in the barrier associated with the critically

constricted state due to the presence of the second membrane in the DM system (Fig. 3a). Thus, the canonical pathway through sequential disconnection of the inner and outer monolayers does not explain the apparent facilitation of inner-tube fission that we experimentally observed for DM tubes.

Unexpectedly, further analysis revealed an alternative, two-intermediate remodeling pathway for the DM system, in which the inner and the outer membrane reversibly hemifuse. Hemifusion (HF) of the inner and outer membranes appears to be metastable in our calculations (Fig. 3b, c and Supplementary Movie 2). Notably, the barrier of the rate-limiting step, starting from the hemifused state, is significantly smaller than the barrier to direct inner-tube fission (Fig. 3b). But even if calculated relative to the unperturbed DM tube, the barrier to fission via the HF pathway is smaller than for direct fission (Fig. 3b). Therefore, our theoretical predictions guide interpretation of the experiment, as less local constriction stress (or less local curvature, as seen in Fig. 1) is required to trigger fission in the DM system.

## Main intermediates of the DM fission pathways

Next, we tested if the metastable contacts between DM tube's inner and outer membranes under moderate constriction can be detected experimentally. As such structures (localized below 10 nm) are well beyond the fluorescence resolution limit, we turned to ultrastructural characterization via cryoEM. We adapted the reservoir "rolling" method described above to form SM and DM tubes directly on holey EM grids (see "Methods", Supplementary Fig. 1c). To observe the fission intermediates, we developed a method of fast-freezing membrane tubes formed on the grid right upon application of the constriction force (hydrodynamic flow, see "Methods", Supplementary Fig. 1c, Supplementary Movie 3). The constriction resulted in tubes' breakage, detectable in the frozen samples (Fig. 2, Supplementary Figs. 2, 3 and Movie 3). As stated above, less curvature stress is needed to break the inner membrane of the DM tube. Accordingly, under similar constriction force (i.e., same hydrodynamic flow), we observed fewer DM tubes as compared to SM tubes (16 DM and 94 SM tubes in 7 independent experiments, see also the distribution in Supplementary Fig. 1b).

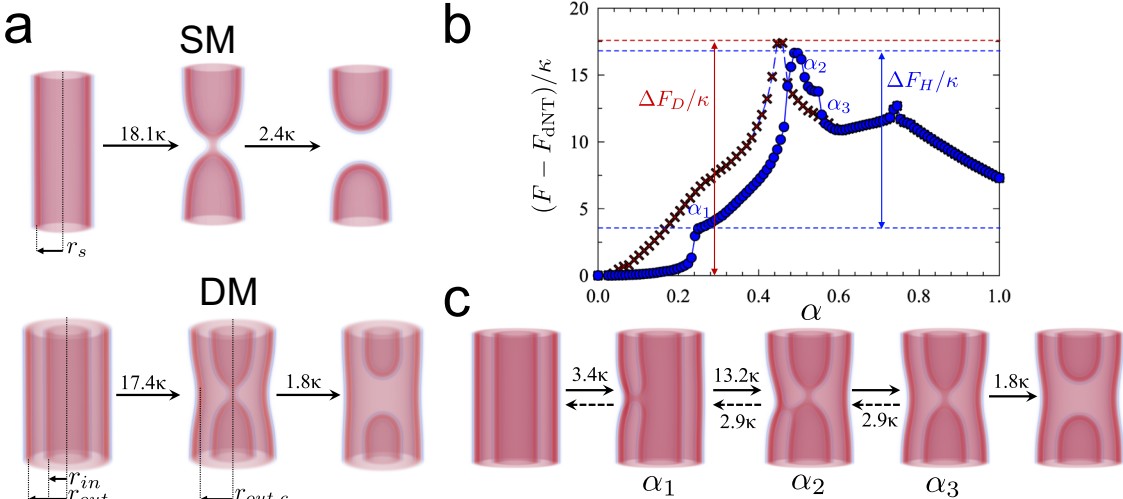

**Fig. 3 | Canonical and HF pathways of inner membrane fission in DM background. a** The canonical SM and DM fission pathways with tension set at -1 Dyn/cm. The single tube has a radius $r_s$ - 8.5 nm. The inner membrane in the DM tube has a radius of $r_{in}$ - 8 nm while the outer membrane has a radius $r_{out}$ - 16 nm. During constriction, the outer radius shrinks to $r_{out,c}$ - 13.5 nm, as the outer tube is no longer expanded by hydration repulsion. Membrane configurations and free-energy barriers were calculated using SCFT. **b** Free energies as a function of the reaction coordinate, $\alpha$, for canonical (red crosses) and HF (blue circles) pathways of inner-tube fission at a membrane tension of -1 Dyn/cm. Each local minimum in the curves corresponds to a metastable state shown in (**c**). **c** The HF pathway for membrane fission in a DM tube. Free energies are given in units of the bending energy, κ. Reaction coordinates, $\alpha$, correspond to values in (**a**). Source data are provided as a Source data file.

In SM tubes we detected the formation of WLM structures just before breakage occurred (18% of SM tubes, Fig. 2a, Supplementary Figs. 1 and 3). While metastable WLMs were found in earlier models of SM fission, their existence has not yet been experimentally confirmed in purely lipid systems. Interestingly, our calculations revealed that the probability of WLM formation depends on the initial membrane tension. At high membrane tension, similar to the conditions of the quasi-dynamical cryoEM experiments, the extension of the WLM becomes favorable (Supplementary Fig. 4). Indeed, we could detect extended WLMs even after fission occurred (7% of SM tubes, "m" in Supplementary Fig. 3), thus confirming the metastability of this structure. However, WLMs become unfavorable at lower tension and break without extending significantly (in 35% of SM tubes, Supplementary Figs. 1 and 4). Therefore, the detection of extended WLMs at lower membrane tension, like that of cellular membranes, is less probable. The experimental observation of WLM in high-tension settings, as predicted by our theoretical calculations, further validates our SCFT approach as a powerful tool for predicting membrane rearrangements during membrane remodeling.

Our SCFT calculations foresaw the existence of metastable IMCs in DM tubes under constriction. CryoEM observations of such tubes revealed places of close membranes' apposition. In 56% of DM tubes we detected localized IMCs, corresponding to places with inter-membrane distance below that of the hydration barrier for membrane fusion (<0.5 nm[50]), once again validating the theoretical predictions (Fig. 2, Supplementary Fig. 2). Although most inner membranes of the DM tubes retracted to the reservoir before sample freezing, delayed fission in a few DM tubes allowed detection of the IMCs and structures resembling HF intermediate (Fig. 2c, left; Supplementary Fig. 2). We also detected fission followed by simultaneous retraction of both membranes of the DM tube (Fig. 2c, right), confirming that both membranes are physically interacting at the initial stages of fission, as was predicted by theory and observed in osmotically driven fission experiments above.

Overall, cryoEM data confirmed the appearance of IMCs and HF-like structures in DM systems, in agreement with theoretical modeling.

## Topological characterization of the HF pathway

As ultrastructural analysis confirmed the metastability of the two critical intermediates in the HF pathway, thus linking theory and experiment and establishing the alternative path as dominant due to the low free-energy barrier, we used SCFT to obtain further insights into the alternative pathway and explore its topological variants. SCFT revealed that the HF connection to the outer membrane facilitates local inner-tube constriction (Fig. 4 (i)) and results in the formation of a transient pore in the inner membrane, close to the HF stalk (Fig. 4 (iii)). Such a pore has been previously observed to accompany membrane fusion[51]. However, instead of promoting full membrane fusion, in the multilayered tubular configuration, the pore exclusively affects the remodeling of the inner membrane, i.e., the membrane with higher curvature stress. First, it closes on one side of the inner tube, resulting in a capped tube connected through a WLM to an open, outer-membrane-connected tube (Fig. 4 (iv)). Then, the pore closes completely, resulting in two capped tubes connected by a WLM, with one of them hemifused to the outer membrane (Fig. 4 (v)). Finally, the HF connection to the outer membrane ruptures, resulting in two capped tubes connected by a WLM encircled by the outer membrane (Fig. 4 (vi)). This HF pathway implies transient lipid mixing between the outer and the inner membranes of the DM tube. Besides, this pathway may result in limited content leakage from the inner membrane compartment due to the presence of the transient pore.

The initial barrier in both, the canonical and the alternative HF pathways is linked to the constriction of the inner membrane. At extremely high constriction (radius of the inner tube ≲6 nm) the canonical fission mechanism becomes preferable (see Supplementary Figs. 5 and 6). For the DM systems considered thus far (inner radius 8 nm), the local constriction required for fission via the canonical pathway is larger; thus, the alternative leaky HF pathway is preferred. Therefore, the presence of the second membrane facilitates constriction by an external force (including protein machinery), but simultaneously results in a bias towards a leaky fission. Such an interplay between assistance and leakiness, seen already at the level of pure lipid membranes, might have interesting biological implications.

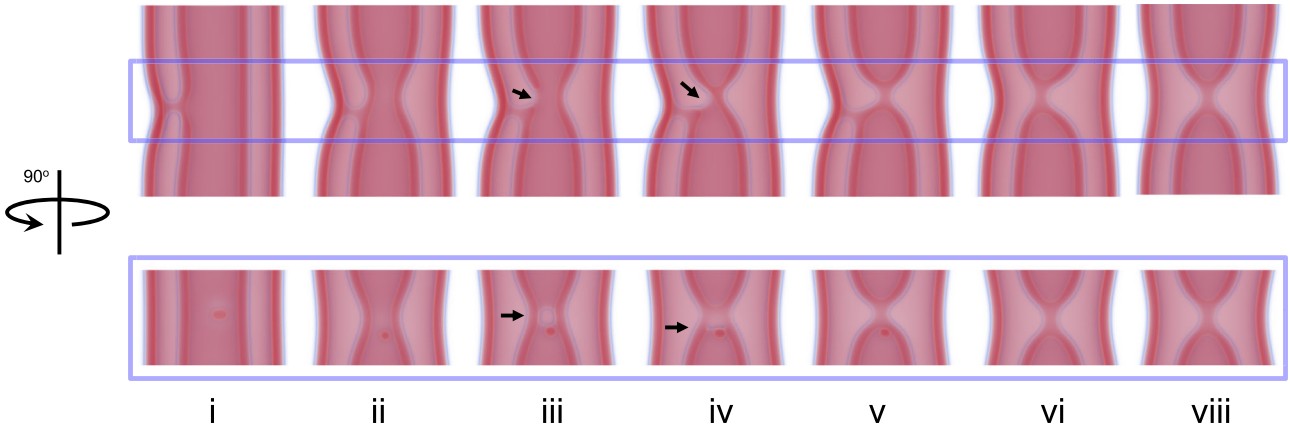

**Fig. 4 | Intermediates in the HF pathway.** The rate-limiting step in the HF pathway to inner tube fission at -1 Dyn/cm. Upper and lower images show views rotated by 90°. Arrows point to the transient pore. See also Supplementary Movie 2.

## Discussion

Formation of a new membrane compartment ultimately requires separation and scission of a SM neck. In a DM system the topological task is more difficult, as the neck has two layers to be cut. DM necks, product of localized constriction of DM organelles and membrane systems (such as ER-organelle contact sites), have been seen in multiple cellular processes. Our work intends to answer one of the open and perhaps most fundamental questions about the pathways and energies of the corresponding lipid rearrangements. Knowing those pathways has been fundamental for our mechanistic understanding of both cellular membrane fusion and fission[11–13,16,18,19,26–32] processes.

As we have learned from decades of studies on SM fission (and fusion), lipids both set the barriers to be overcome by proteins and play an important regulatory role in remodeling. Lipids affect the reaction along the whole path via their elasticity, polymorphism, and ability to form non-bilayer structures and intra- or inter-membrane contacts. All these inherent lipid properties become relevant under constriction stress, the major driver of SM fission in all cellular systems.

Here we reveal an even more complex and interesting behavior of lipids in DM system under constriction stress, giving rise to multiple pathways of rearrangements and topological complexity via formation of metastable IMCs, physical interaction and, most importantly, regulated membrane leakiness. Altogether, the complex lipid behavior in the DM system undergoing fission is fundamental for our understanding of the design and protein action of molecular machineries behind the process. Our work further indicates that the speed and the strength of membrane constriction may affect the extent of inter-membrane contacts during DM fission, as further discussed below.

We emulate membrane constriction produced by specialized proteins by using purely lipidic membranes under tension. Inclusion of proteins in the system may affect the landscape of the reaction in two ways. First, proteins embedded in the membrane may alter the membrane's bending rigidity, making the process occur more or less readily. Second, specialized fission-protein machineries are designed to catalyze the remodeling pathways, making the free-energy barriers of lipid systems more accessible. However, in either case, the primary determinant of the remodeling pathway itself is the lipid membrane. Therefore, our work elucidates the fundamental mechanisms for double membrane fission.

We show that including one more membrane layer yields a new dimension of the fission reaction, with additional options and control parameters to consider compared to single-membrane (SM) systems. While our data indicate that the metastable worm-like micelle (WLM) structure is the key intermediate in all fission pathways, the extra membrane provides various routes towards this intermediate, depending on membrane stress. As expected[10,11], the WLM forms upon

critical constriction in DM fission, thus requiring a high free energy per area that can be supplied by specialized protein machineries.

However, a second membrane enables lower free-energy pathway(s) via a delicate dynamic interplay between the HF of the inner and outer membranes and WLM formation. The barrier decrease for this pathway is more than several $k_BTs$. Thus, the HF mechanism becomes the dominant pathway, with the extra membrane acting as leverage, lowering the force requirements to initiate the inner-membrane fission. The free-energy barrier difference becomes even bigger (becoming half of the canonical one) if normalized by units of the circumference of the dividing tube. Furthermore, our data indicate that the tradeoff between the canonical and the HF pathway depends on the initial curvature of the dividing system. Similar to our theoretical system, in cells, such control of the initial membrane constriction can be provided by membrane tension, both intrinsic to each membrane or caused by friction, and local cytoskeleton or motor protein action, shown to be of crucial importance for organelle division[20,52,53].

Despite being energetically favorable at initial constriction radii, the HF pathway to fission may be a leaky one. A small and short-lived pore forms in the inner membrane before the WLM emerges, allowing for local loss in membrane asymmetry and transient content leakage. Such a pore may have a physiological role in fission, precisely triggering localized ion or lipid composition changes at the fission site. It also underlines the importance of the protein machinery in this process. Less powerful constriction machines, such as the mitochondrial fission protein Drp1[54,55], may lead to a slightly leaky pathway where both membranes contact each other. Other, more powerful machineries, such as the isoform 2 of the classical dynamins (Dyn2), ubiquitous in the cell[56], may lead to the canonical sequential fission of the double membrane.

Interestingly, Drp1 is an inefficient membrane constrictor, incapable of mediating membrane scission of SM in vitro[54,55], whereas Dyn2 exhibits enhanced curvature sensing, assembling into constriction units only on moderately curved membranes[57]. As proposed earlier, both types of constrictors may work in synergy[58], leading the DM system first through the HF pathway of inner membrane fission and then completing the outer membrane scission through the canonical pathway. Finally, our study opens the possibility for inner-membrane fission without the requirements of high membrane constriction and further fission of the outer membrane. This process can be related to ER wrapping at organelles' fission sites[4,5].

The HF pathway implies direct contact between the inner and outer membranes of the DM system through the HF diaphragm. In cells, both membranes are usually set apart by intermembrane protein spacers. However, ultrastructural observations of mitochondrial fission necks suggest that transient contacts between constricted

mitochondrial membranes are possible due to their proximity[59–62]. While yet to be confirmed experimentally, such brief intermembrane connections could also exist at the edges of the contact sites between the ER and the dividing organelles. Besides helping in WLM formation, the HF contacts would facilitate a fast interchange of lipid moieties between both membranes, a process postulated to happen at the fission sites of DM systems[63]. Therefore, the HF pathway for DM fission may lead to the formation of purely lipidic contact sites between the membranes.

Though further theory and experiments on lipid dependence and protein involvement are needed, our study reveals a much larger landscape of membrane fission pathways and mechanisms in multilayered systems. Membrane wrapping brings in more than force help; it enables several remodeling pathways, with constriction and leakiness being their major determinants. These findings have clear implications for the role of ER at the fission site of different organelles, establish additional links between mitochondrial fission and apoptosis, and bring essential information for further studies of DM fission events relevant to cellular homeostasis.

## Methods
### Osmotically induced fission of lipid nanotubes
Fluorescently labeled lipid nanotubes were produced by mechanical "rolling" of 40 μm silica beads covered by hydrated membrane lamella on top of a SU8 micropillar array, as previously described[47] (Supplementary Fig. 1a). Freely suspended SM and DM tubes form between the reservoirs deposited on top of the pillars (Supplementary Fig. 1a). The lipid nanotubes were prepared in a hypotonic solution containing 1 mM Hepes pH 7.0. and were monitored from beneath using an inverted fluorescence microscope (Nikon Eclipse Ti, Japan) equipped with a 100X/1.49NA objective lens, a CoolLed pE-4000 light source, and a Zyla 4.2 sCMOS camera (Andor, Ireland). Nanotube fission was induced by perfusion of the experimental chamber with a hyperosmotic solution containing 1 M trehalose in the presence of 1 mM Hepes, pH 7.0. μManager software (version v1.4.22) was used for image acquisition[64]. Image processing (background subtraction and kymograph building) and statistical analysis were performed using Fiji package in ImageJ[65] (version 2.14) and Origin (version 8.0 or 2020) software, respectively.

### Fluorescence microscopy-based characterization of tubes' geometry and topology
Tubes' geometries were quantified as previously described, from fluorescence intensity calibration of the lipid film on a flat surface[47,48].

Briefly, a flat, supported lipid bilayer (SLB) was formed on a clean, plasma-treated cover glass by bursting of giant vesicles made with the same lipid composition as the nanotubes. The SLBs were used to find the density of the membrane fluorescence signal ($D$), defined as the total fluorescence per membrane area. The nanotube radius was obtained from the total fluorescence per unit length of the nanotube, $F_l$, using $r = F_l/2\pi D$. Note that the radius refers to the distance between the center of the tube and the middle point of the bilayer. For DM tubes, the inner membrane tube radius was estimated by assuming a constant intermembrane space between the lipid head region of the outer monolayer of the inner tube and the inner monolayer of the outer tube of 2 or 5 nm, respectively. Bilayer thickness was assumed to be 5 nm (Fig. 2b, Supplementary Fig. 1e).

The lamellarity of the tubes was determined by plotting the distribution of the areas under the fluorescence intensity peaks normal to the tube axis[47] (Supplementary Fig. 1b).

### Formation of lipid tubes for cryoEM characterization
DM and SM tubes were produced directly on the glow-discharged Quantifoil R 2/2 300 mesh copper grid. A 3 μL drop of 1 mM Hepes, pH 7.5 solution was placed on the grid. Then, 40 μm beads covered with lipid lamella (containing 1-hexadecanoyl-2-(9Z-octadecenoyl)-sn-glycero-3-phosphocholine (POPC) and 1,2-dioleoyl-sn-glycero-3-phosphoethanolamine-N-(lissamine rhodamine B sulfonyl) (ammonium salt) (RhPE) at 99:1 mol%, 1 g/L) were introduced into the liquid drop and slowly rolled on the grid surface by tilting the grid. The grid was then placed on a Thermofischer Vitrobot system, maintained at 18 °C and 90% humidity. The liquid was removed by blotting with an absorbent filter paper on both sides of the grid for 2 s. The grid was abruptly plunged into liquid ethane (−184 °C).

For the quasi-dynamical experiments, the samples were prepared as follows. POPC and RhPE were mixed from the corresponding stocks in chloroform to a final 99:1 mol% and 1 g/L concentration. Chloroform was evaporated under vacuum, and the resulting lamellae were rehydrated in 1 mM Hepes pH 7.0. 0.4 μL drop of the final multilamellar vesicle solution was placed on the edge of a holey carbon-coated side of a glow-discharged Quantifoil R 2/2 300 mesh copper grid and dried under vacuum for 30 min. The grid was picked up with forceps near the lipid reservoir and was placed inside the chamber of a Leica EM GP2 cryo-plunger (Leica®), maintained at 15 °C and 90% relative humidity. Next, 3 μL of 1 mM Hepes pH 7.0 were added to the grid from the carbon-coated side. The lipids were allowed to rehydrate for 2 min. After 2 min of incubation, the liquid was removed by blotting with an absorbent filter paper (Ø55 mm, Grade 595, Hahnemühle) for 3 s on the grid edge opposite to the lipid reservoir. Following, the grid was abruptly plunged into liquid ethane (−184 °C) (Supplementary Fig. 1c).

### CryoEM observation of lipid tubes
The vitrified samples were maintained in liquid nitrogen and visualized on a JEOL JEM-2200FS/CR, equipped with a field emission gun (FEG) operated at 200 kV and an in-column Ω energy filter or a ThermoFisher Titan Krios G4 Cryo-TEM, equipped with an extreme high-brightness field emission gun (X-FEG) operated at 300 kV and Gatan BioContinuum Energy Filter. During imaging, non-tilted zero-loss two-dimensional (2D) images were recorded under low-dose conditions, utilizing the 'Minimum Dose System (MDS)' of Jeol software or EPU software, with a total dose on the order of 30–40 electrons/Å² per exposure, at defocus values ranging from 1.5 to 4.0 μm. The microscope's in-column Omega energy filter helped us record images with an improved signal-to-noise ratio (SNR) by zero-loss filtering, using an energy-selecting slit width of 20 eV centered at the zero-loss peak of the energy spectra. For JEOL JEM-2200FS/CR, digital images were recorded in linear mode on a Gatan K2 Summit direct detection camera (Gatan Inc.) (5 μm/pixel size, 3840 × 3712 pixels). For the in the ThermoFisher Titan Krios system, images were acquired on a Gatan K3 direct detection camera (Gatan Inc) (5 μm/pixel, 5760 × 4092 pixels). In both cases, DigitalMicrograph™ (Gatan Inc.) software was used at nominal magnifications of 2000× and 25,000× with a pixel size of 1.6 nm and 0.154 nm, respectively (JEOL JEM), and 42,000× with a pixel size of 0.42 nm (ThermoFisher Titan Krios). Images were subsequently treated and analyzed using Fiji package of ImageJ software[65]. Statistical analysis was performed using Origin (version 8.0 or 2020) software. Images were assayed for IMCs by applying a two independent sample t-test (equal variance assumed) to confirm the alternative hypothesis [(mean DM thickness)/2 − mean SM thickness <0.5 nm] at 0.05 significance level.

### Self-consistent field theory
Biological bilayer membranes are formed by the self-assembly of lipid molecules in water. The lipids are amphiphilic, composed of a hydrophilic head group and a hydrophobic tail group. We represent a lipid as an AB diblock copolymer composed of $fN$ segments of type A (tail) and $(1-f)N$ of type B (head). In this work we fix $f = 0.8$. The solvent, typically water, is represented as a short homopolymer composed of $N_s = N/10$ segments of B type. The molecules are modeled as Gaussian chains, where each segment has a statistical length, $b$, and volume $\rho^{-1}$.

The natural end-to-end distance of a lipid (copolymer) is thus $R_0 = b\sqrt{N}$, which we use as our unit length.

The repulsion between A and B segments is characterized by the Flory-Huggins interactions parameter, which we fix at $\chi N = 30$. Calculations are done in the semi-grand canonical ensemble. The numbers of each chemical species, $n_l$, and $n_s$, are not fixed. Rather, we fix the overall average monomer density, $\rho$, and the exchange chemical potential, $\mu = \mu_l - \mu_s$, which is the difference between the chemical potentials of lipids and solvent molecules. This chemical potential can be related to the membrane tension, $\sigma$, which we use to characterize the membrane. The equilibrium behavior of our lipid solution is calculated by SCFT of the standard model of Gaussian chains[24,25]. Further details are given in the Supplementary Information.

For a simple transition one can often define a reaction coordinate that characterizes the reversible transformation pathway and calculate the free energy as a function thereof, however, complicated membrane rearrangements offer no simple, reversible reaction coordinate. In order to study membrane transformations, we use the string method[37,38,66–69] and define a reaction coordinate given by the root-mean-squared change in local density throughout the transformation of membrane topology. The string method allows us to find the MFEP connecting configurations[37,38] and thus a thermodynamic estimate for the most probable transformation pathway between states.

### Reporting summary

Further information on research design is available in the Nature Portfolio Reporting Summary linked to this article.

## Data availability

Source data underlying the Fig. 1c, and Supplementary Fig. 1b and e are provided as a Source data file. Raw images for Fig. 2 are shown in Supplementary Figs. 2 and 3. Statistical analysis presented in Fig. 2 is provided as Source data file. The SCFT code is available on Github [https://github.com/PolymerTheory/SCFT_string]. Raw data has been deposited to Figshare [https://doi.org/10.6084/m9.figshare.25407907]. Source data are provided with this paper.

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

## Acknowledgements

We thank Vadim Frolov for many helpful discussions and David Gil Carton for technical advice. Financial support has been provided by the Deutsch Forschungsgemeinschaft within CRC 1286 TP C06. A.V.S. was supported by the Grant PGC2018-099971-B-I00 and PID2021-127844NB-I00 funded by MCINAEI 10.13039501100011033 and by "ERDF A way of making Europe", by the "European Union" and by the Basque Govern-ment Grant IT1625-22. The authors gratefully acknowledge the Gauss Centre for Supercomputing e.V. (www.gauss-centre.eu) for funding this research project by providing computing time through the John von Neumann Institute for Computing (NIC) on GCS Supercomputer JUWELS at the Jülich Supercomputing Centre (JSC). The authors are grateful to the Electron Microscopy and Crystallography platform of the CIC bioGUNE and the Basque Resource for Electron Microscopy for providing access to cryo EM sample preparation and analysis equipment.

## Author contributions

Conceptualization: R.K.W.S., A.V.S. and M.M.; experimental methodology: A.V.S.; fluorescence microscopy-based fission assays and analysis: J.M.M.G. and A.V.S.; cryoEM based quasi-dynamic fission assay: I.S.P., I.R.R. and A.V.S.; cryoEM data analysis: A.V.S.; theoretical methodology and analysis: R.K.W.S. and M.M.; writing—original draft: R.K.W.S., A.V.S. and M.M; writing—review & editing: all authors; funding acquisition: A.V.S. and M.M.; resources: M.M. and A.V.S.; supervision: M.M. and A.V.S.

## Funding

## Competing interests

The authors declare no competing interests.
