## [Peer Review File · Nature Communications]

Membrane fission via transmembrane contactREVIEWER COMMENTS

Reviewer #1 (Remarks to the Author):

Summary:

This manuscript investigates mechanisms mediating the fission of double membrane cellular systems (i.e., organelles). The authors develop an in vitro approach to reconstitute a double-membrane-tubular system and examine the effect on the stability of both membranes upon varying degrees of stress (i.e., osmotic stress or membrane tension) using light microscopy and cryo-electron microscopy. They analyzed the experimental data using a self-consistent field theory (SCFT) model, representing the first time this type of analysis has been applied to investigate complex, double membrane systems. The authors find that the double membrane's presence seems to influence the second membrane's fission, suggesting that there is likely a transient intermembrane contact that mediates a favorable pathway for the fission of the inner membrane.

Significance:

As a reviewer with a strong background in using cryo-electron microscopy to study organellar dynamics and ultrastructure, my primary focus is to evaluate these aspects of the manuscript. One of the manuscript's strengths is developing a new method to study double membrane systems by cryo-EM shortly after a force is applied via hydrodynamic flow. Furthermore, applying the SCFT model to analyze potential membrane constriction pathways is also new and could potentially benefit the broader fields of organellar dynamics and membrane biology.

Suggestions for improvement

1. One of the most significant areas of improvement the authors need to address is the applicability of their experimental results/conclusions to physiologically relevant cellular membranes. For example, their experimental and computational models assume organellar membranes are composed entirely of lipids and devoid of proteins. We know this is not true for organellar membranes, which are often composed almost entirely of proteins. For example, the outer membrane of mitochondria contains a 50-to-50 protein-to-lipid ratio, and the inner membrane exhibits an even higher number of proteins, with an estimated ratio of 80-to-20. It is hard to imagine that this high level of proteins would not substantially influence the stability, dynamics, and structure of double membrane organelles upon force application.

Given that all the experimental and computational modeling data is performed without proteins, it is difficult to contextualize the results of these experiments to physiologically relevant systems. I believe the modeling results presented help inspire potential hypotheses and models in the organellar dynamics field for how double membrane fission may proceed; however, extreme caution and consideration should be taken when directly comparing these results to organelles like mitochondria. Most of the manuscript is framed in a way that implies these calculations and results could be readily applied to

endoplasmic reticulum-driven mitochondria fission, which I believe could be problematic for the field given that these do not consider protein content within the membrane. A potential path forward would be to either: (1) incorporate membrane proteins into the experimental and modeling experiments or (2) adapt the manuscript's language to ensure that the results are interpreted appropriately (e.g., make parallels to in vitro systems and specific cellular membranes with very low protein to lipid ratios).

2. Similarly, it would be helpful for the authors to provide more biological contexts for the types of forces/tensions/osmotic stresses they use in their experiments and simulations. How does this compare to other relevant force-generating proteins or organelles involved in organellar membrane constriction? For example, the authors state: "Our calculations indicate that at a relatively high but physiologically relevant tension 1 Dyn/cm"—to what biological process with comparable tension can this be related?

3. The statement made in line 223 ("However, the barrier of the canonical pathway...") further calls into question the relevance of their findings since we know most fission of mitochondrial membranes is indeed controlled by cytoplasmic protein constriction machinery. In what contexts do the authors deem their findings relevant?

4. Most of the claims about the relevance of their simulation analysis come from only a few cryo-EM projection images. From these images alone, it is difficult for the reader to evaluate the presence of HF or WLM states (model shown in 3B). Even for this reviewer with a trained cryoEM "eye," it is challenging to discern what is happening in figure 3C and D. This is concerning because their modeling analysis' biological relevance hinges on detecting these intermediate states experimentally via this cryoEM data. Quantifications and/or more examples of cryoEM images (e.g., an image gallery) are necessary to further support the claims made from this data (e.g., the existence of HF or WLM states). Additionally, more annotations on the images themselves will help with the interpretability of low signal-to-noise cryo-EM images to a broader audience. I also encourage the authors to increase the size of all their figures, especially those that contain microscopy images.

5. The micrograph presented in Supplementary Figure appears to have contamination (i.e., "leopard" ice), further complicating the findings' interpretability.

6. I am not following the logic of these two statements; could the authors please clarify? "Remarkably, we found only a slight decrease in the barrier associated with the critically-constricted state due to the presence of the second membrane in the DM system (Fig. 2). Thus, the canonical pathway through sequential disconnection of the inner and outer monolayers does not explain the apparent facilitation of inner-tube fission that we experimentally observed for DM tubes."

7. There is a lack of technical or experimental details on how the images are generated from the modeling analysis in Figures 2-4. Additional details are necessary to describe how these graphical representations of the modeling analysis are generated.

8. In line 247: "The barrier decrease for this pathway is significant."—how is significance determined in this case?

9. In line 278: “However, ultrastructural observations of mitochondrial fission necks suggest that transient contacts between constricted mitochondrial membranes are possible due to their close proximity^{16–18,64}.” Some of these referenced studies show that mitochondrial membranes constrict simultaneously, even in the most constricted states, which contradicts the interpretation/statement above. Could the authors comment on this further?

Reviewer #2 (Remarks to the Author):

The growth and division of cellular organelles frequently involves wrapping of additional or multiple membranes. How the presence of additional membrane affects the process of division remains unanswered in the field. In this manuscript, authors have identified the instabilities generated by the stress due to the presence of an extra membrane in a double-membrane system. Using Fluorescence, cryo-EM and Self-consistent field theory (SCFT), authors have studied the membrane remodeling pathways during fission, generated due to osmotic stress or membrane tension. The intermediate states during membrane fission like worm-like micelle structure and hemifusion state were also studied.

The article seems to be interesting and in principle it would be a good addition to the field. Although below are few major concerns and comments authors should address to improve the manuscript.

- Rapid changes in osmolarity, resulting from water flow, should induce membrane bulging as water moves inside a closed membrane system. An important question to address is whether this membrane expansion contributes to the fission process. Authors should elaborate on this point to provide a more comprehensive understanding of the phenomenon.
- In Fig. 1a, authors should elaborate more on how they differentiated between a single, double or multi-membrane system in the fluorescence experiments? Is it possible to mention the change in tube diameters or change in fluorescence intensities in the text? How the changes between DM and multi-membrane systems can be distinguished? In Fig. 1b legends, does n=3 represent the number of tubes or on the number of times experiment was performed?
- It's not clear how the authors distinguish between inner membrane fission versus outer membrane fission based on fluorescence microscopy. This should be described in more detail. Also, the result that the inner membrane undergoes fission before the outer membrane should be supported by cryoEM images showing an intermediate state with the inner membrane separated and the outer membrane intact.
- In Fig. 2, can the measurements on outer membrane diameters be provided? Does it remain the same or change during constriction? A comparison of outer diameter of SM and DM would be beneficial.
- Estimating the exact dimensions of nanotubes requires a single supported lipid bilayer (SLB) [1].

However, this can be challenging to compute when authors employ microsphere beads to create nanotubes, as these microbeads can introduce excess membrane, complicating the analysis. Therefore, the authors should consider determining tube dimensions during water-induced membrane fission using cryo-electron microscopy (cryoEM).

- Starting tube diameter also influences membrane fission based on the protein-mediated fission literature. In the double membrane system, the inner tube would have higher curvature stress and so may undergo fission preferentially. Are the tube diameters for the inner double membrane tube and single membrane tube (figure 1a) similar? These should be the same to conclude that inner double membrane tube undergoes preferential fission. The authors should consider analyzing the dependence on tube diameter to support their conclusion that the inner membrane undergoes preferential fission.

- Line 179 Panel 2c is missing from Figure 2.

- The current result on the formation of worm-like micelle under high tension seems qualitative as tension cannot be measured in the current setup. Is there a way to quantitate the threshold tension over which WLM is readily observed? A quantification of the appearance WLM structure in cryo-EM data would help. The authors should compare tubes formed with bead rolling versus tubes formed with blotting-induced fluid flux by cryoEM if possible.

- Can the system be tested with the addition of proteins that cause membrane fission like dynamin? It would be interesting to know how the system behaves in more physiologically relevant conditions.

1. Dar S, Kamerkar SC, Pucadyil TJ. Use of the supported membrane tube assay system for real-time analysis of membrane fission reactions. *Nat Protoc* 2017; 12: 390–400.

Reviewer #3 (Remarks to the Author):

The authors interrogated the fission event in both single membrane (SM) and double membrane (DM) systems via experiments and self-consistent field theory (SCFT). The authors concluded that while the worm-like micelle (WLM) is the critical intermediate shared by both SM and DM fission pathways, the outer membrane in the DM systems can transiently contact the inner membrane and form the hemi-fusion structure (HF) which can lower the fission energy but result in leakiness. The paper is well-written and organized. While the finding sounds exciting and attractive and provides a new insight for the DM fusion pathway, however, more justification for the method, calculations and experimental validation need be provided to support their claims.

1) As claimed by the authors on line 85 and 86, the SCFT are mainly applied to small scale membrane

transformations in the past, but the scale of magnitude of their simulation system is much larger. As a result, is the method still sound and rigorous in the current system? What is the limitation of the SCFT method? And what are the assumptions of this theoretical treatment? To validate the theoretical method, one typically needs to use the precise experimental measurements as a benchmark. In this regard, what is the experimental benchmark in the current system? And can the authors provide any error estimation of their theoretical method?

2) The authors claimed that the HF pathway breaks the canonical pathway into several intermediate states. This way, the transition between the intermediate states requires less energy, lowering the energy barrier of the rate-limiting steps and as well as the overall energy. Does this energy reduction come from entropy or enthalpy? In Fig 3 a, the energy at $\alpha=0$ (fission not started) is lower than $\alpha=1$ (fission completed). Is this attributed to the additional membrane-water interface created by the fission? To provide a clear physical picture of this noncanonical pathway of membrane fission, the authors need to specify the different sources of energy change corresponding to different stages of the fission.

3) For the hemi-fusion pathway, the pore formation would expose the hydrophobic tails of the lipid, which is energetically unfavorable in general. How was this energy penalty compensated in the model? One possibility is that the formation of the pore promotes the flip-flop of the lipids, which is not depicted in the model. This leads to a more fundamental question of the lipid exchange during the fission pathway. The authors need to either incorporate this effect in their theoretical model or prove that the flip-flop/lipid exchanges do not notably contribute to the fission of double membrane.

4) In a broader scope, the main conclusion of this work stems from theoretical predictions. The problem is: Both the theory and the calculation in the work are built upon several assumptions of the membrane shape and dynamics, which are not first-principle. To substantiate their conclusion, the authors need to experimentally validate either the assumption or the prediction of the theoretical model, or both. As it currently stands, it is difficult to appreciate exactly how experiments validate the essence of the model. For instance, what is the statistics of cryoEM data in Fig 3c-d? It is just one particular image, or a general phenomenon. If the latter, then what is the sample size, and what is the percentage of the cases that clearly shows the fission/the inner-outer membrane contact? What is the proper experimental control for Fig. 3c-d? How can we be sure that the cryoEM experimental procedure does not introduce any perturbation on the double membrane shape? Without the rigorous experimental validations, the conclusion can only be viewed as a hypothesis, which greatly decreases the significance of the work.

Point-to-point response to reviewer comments

We thank the reviewers for their constructive feedback on our manuscript. Below we provide a point-by-point response to each of the reviewer's concerns, whose comments we reproduce italicized in bold. Our responses follow.

Reviewer 1:

1. *Given that all the experimental and computational modeling data is performed without proteins, it is difficult to contextualize the results of these experiments to physiologically relevant systems. I believe the modeling results presented help inspire potential hypotheses and models in the organellar dynamics field for how double membrane fission may proceed; however, extreme caution and consideration should be taken when directly comparing these results to organelles like mitochondria. Most of the manuscript is framed in a way that implies these calculations and results could be readily applied to endoplasmic reticulum-driven mitochondria fission, which I believe could be problematic for the field given that these do not consider protein content within the membrane. A potential path forward would be to either: (1) incorporate membrane proteins into the experimental and modeling experiments or (2) adapt the manuscript's language to ensure that the results are interpreted appropriately (e.g., make parallels to in vitro systems and specific cellular membranes with very low protein to lipid ratios).*

We thank the reviewer for the suggestions. We are afraid that the first approach is not feasible within the scope of the present manuscript; yet it is a valuable suggestion for future work. Hence, we rewrote the manuscript to emphasize that we are not modeling a concrete biological process but instead are concerned with the basic, yet fundamental, physics of double-membrane fission.

We are fully aware of proteins being highly present in any biological membrane, where they have two clearly distinguishable roles:

1. The proteins generally contained within the membrane modify the mechanical properties of the membrane (such as bending rigidity and spontaneous curvature e.g., by flipping lipids between monolayers or altering the local membrane composition). However, this is not expected to qualitatively affect the remodelling pathways available to the membrane. It is typical for research on membrane behavior to focus on the phospholipid bilayer, without reference to these passive effects of membrane proteins¹⁻⁴. Even research conducted on the active role of proteins (role 2, below) typically considers the protein machinery on the phospholipid bilayer without specific reference to the passive effects (such as e.g., changes in mechanical properties)^{5,6}.
2. Proteins specifically involved in membrane remodelling may have a catalytic effect on the remodelling pathway by lowering the free-energy barriers at different steps of the transformation path. E.g., proteins involved in membrane fusion may (i) bring the "right" membranes together (recognition), (ii) dehydrating the membrane contact, (iii) imparting tension onto the membrane, and (iv) exerting forces on the hemifusion diaphragm, setting the stress and shape of the lipid membrane. Importantly, however, while proteins may decrease the free-energy barriers, the topological remodeling pathway itself (secluded to just a few nm and ms) is dictated by the membrane lipid matrix.

The role of fission proteins is similar for the case of double membrane fission. The fission machinery (e.g., dynamins) constrict the tube (fission neck), increasing its curvature stress. In our first study of double membrane fission, we simulate this constriction by considering tubes under tension, as the radius of a membrane tube is inversely proportional to the square root of the membrane tension⁷ (see the equation below for details). Our goal is to demonstrate that the new fission-by-fusion pathway is a viable mechanism with barriers similar to or smaller than the classical pathway and that it can help explain discrepancies in fission rates that we observe experimentally.

We have altered the text (see final introduction paragraph and discussion section) to clarify that we mimic protein constriction by tension and that the propensity that we predict for each mechanism presented may be altered by specialized protein machineries. Incorporation of the “protein factor” may shift the boundaries between parameter regions, making one or the other mechanism preferred, but we do not expect qualitative changes in the pathways themselves by virtues of the universality of topological changes in membrane systems.

2. Similarly, it would be helpful for the authors to provide more biological contexts for the types of forces/tensions/osmotic stresses they use in their experiments and simulations. How does this compare to other relevant force-generating proteins or organelles involved in organellar membrane constriction? For example, the authors state: “Our calculations indicate that at a relatively high but physiologically relevant tension 1 Dyn/cm)”—to what biological process with comparable tension can this be related?

Tension and osmotic stress are merely used here as prototypical (and simplest) constriction forces. It is well established in the field that collective lipid behavior/membrane changes can be studied regardless of the constriction force, the final curvature being the only important, relevant determinant. Note that the tension of 1 Dyn/cm creates less curvature than that produced by fission proteins and hence it is not critically high.

In the SCFT calculations, the membrane tension, σ , is measured in units of bending rigidity, κ , and bilayer thickness, d . This dimensionless characteristic is directly related to the experimentally accessible radius, r_{NT} , of a membrane tube via

$$\frac{\sigma d^2}{\kappa} = \frac{1}{2} \left(\frac{d}{r_{NT}} \right)^2$$

We also note that, at the tension considered, the free-energy barriers along the different pathways are smaller than the barriers of pore formation, according to classical nucleation theory.

3. The statement made in line 223 (“However, the barrier of the canonical pathway...”) further calls into question the relevance of their findings since we know most fission of mitochondrial membranes is indeed controlled by cytoplasmic protein constriction machinery. In what contexts do the authors deem their findings relevant?

Again, we do not claim that mitochondria fission is driven exclusively by tension. Yet, based upon earlier analyses of membrane fission, we postulate that the behavior (instability) of the double-membrane lipid system under tension-driven constriction shall be qualitatively similar to that under protein-driven constriction. This is corroborated by SCFT calculations (not shown),

demonstrating that the radius of the constriction indeed is a good reaction coordinate for the initial portion of the pathway. In the present theoretical and experimental work, this constriction is brought about by membrane tension (see equation above), whereas in the biological setting it is created by protein machinery.

In our SCFT calculations and experiments where we induce fission by liquid flow, we mimic the effect of the cytoplasmic protein constriction machinery by an increase of membrane tension. Counterintuitively, the canonical mechanism of sequential fission becomes relevant only at extremely large membrane constriction (see previous response), while the new leaky fission-by-fusion mechanism is relevant for less constricted membranes. Note that the cytosolic protein constriction machinery, mainly Drp1, has been shown to be slow and rather inefficient in its constriction/fission function as compared to the classical dynamins in endocytosis⁸. Therefore, the fission-by-fusion mechanism is especially relevant in the case of mitochondrial fission, where transient intermembrane contacts may be even a requirement for the recruitment of the necessary cofactors for further membrane constriction⁹⁻¹¹. We have clarified this statement in the manuscript now.

4. Most of the claims about the relevance of their simulation analysis come from only a few cryo-EM projection images. From these images alone, it is difficult for the reader to evaluate the presence of HF or WLM states (model shown in 3B). Even for this reviewer with a trained cryoEM “eye,” it is challenging to discern what is happening in figure 3C and D. This is concerning because their modeling analysis’ biological relevance hinges on detecting these intermediate states experimentally via this cryoEM data. Quantifications and/or more examples of cryoEM images (e.g., an image gallery) are necessary to further support the claims made from this data (e.g., the existence of HF or WLM states). Additionally, more annotations on the images themselves will help with the interpretability of low signal-to-noise cryo-EM images to a broader audience. I also encourage the authors to increase the size of all their figures, especially those that contain microscopy images.

We are grateful to the reviewer for this suggestion. We have now provided the Supplementary Movie 3 that supports that cryoEM data presented in Fig.3 by showing nanotube fission induced by hydrodynamic flow observed by fluorescence microscopy. We have also included additional micrographs that support the existence of metastable WLMs upon NT fission in the single membrane systems (Supplementary Fig. 3), as well as cartoons that should help clarifying the data in the panels, suggested by the reviewer.

We note that detection of transient intermediates of membrane remodelling by cryoEM has never been attempted before and is indeed methodologically challenging. However, our dataset allows us to support the major points of the theoretical predictions. We note that the method presented here enables trapping and observing metastable states in a dynamical process far from equilibrium, otherwise inaccessible by other experimental technique, to the best of our knowledge.

Finally, we would like to point out that the relevance of our findings does not “come from only a few cryo-EM projection images”, as stated by the reviewer, as the SCFT has proven to be a very reliable tool in predicting the structure and thermodynamics of amphiphilic self-assembly and has been successfully applied to membrane fusion. Therefore, we think that the techniques

used in this study represent a very powerful toolbox for the mechanistic dissection of dynamic membrane remodelling during fission.

5. The micrograph presented in Supplementary Figure appears to have contamination (i.e., "leopard" ice), further complicating the findings' interpretability.

While we agree that “leopard” ice indicates irregularities of not well understood origin in the ice surface, it is unclear to us how it affects data interpretation. In this case, the lipid WLMs, which are several tens of nanometers long and a few nanometers thin, are clearly visible and quantifiable, rendering data in agreement with the predicted thickness of the WLMs. We also note that protein samples in “leopard” ice have been used for structural analysis (e.g., Ref 12). However, we appreciate the suggestion for improvement, and have now included more images showing the formation of the WLM to support our claim.

6 I am not following the logic of these two statements; could the authors please clarify? “Remarkably, we found only a slight decrease in the barrier associated with the critically-constricted state due to the presence of the second membrane in the DM system (Fig. 2). Thus, the canonical pathway through sequential disconnection of the inner and outer monolayers does not explain the apparent facilitation of inner-tube fission that we experimentally observed for DM tubes.”

We evaluated the barrier to fission via the canonical pathway for (i) a single tube and (ii) for the inner tube in a double-membrane system under same tension/constriction. The presence of the outer membrane has a negligible effect^a on the fission pathway of the inner tube. Therefore, the energy barrier for the fission of the inner tube of the double-membrane system is comparable to that of the single membrane system. In other words, if the canonical fission mechanism is considered, fission of the inner tube in the double-membrane system should proceed at the same rate as the fission of in the single-tube system.

However, this prediction disagrees with the experimental outcome, where the inner tube in the double-membrane system requires less energy to undergo fission, i.e. the fission probability increases. Such result implies the existence of an alternative mechanism for the fission of the inner tube, different from the canonical pathway. We have modified the text to clarify this point.

7. There is a lack of technical or experimental details on how the images are generated from the modeling analysis in Figures 2-4. Additional details are necessary to describe how these graphical representations of the modeling analysis are generated.

^a There is a small decrease in the barrier, which can be attributed to the constriction of the inner tube by virtue of it being squeezed by the outer tube. The decrease in barrier is consistent with the decrease that we obtain from increasing the tension in a single tube to match the radius of the inner tube in the double-membrane system. I.e., when we increase the tension so that the radius of the single tube shrinks from 8.5 nm to the radius of the inner tube in the double-membrane system, 8 nm (now shown in Fig 2), the barrier decreases from approx. 20 κ to 18 κ , based on recalculation at higher tension. This is approximately the same as the value for the double tube via the canonical pathway. The important point is that the double-tube constriction effect on the barrier is small when compared with the difference in barrier between the classical and new pathways.

SCFT calculations produce a field representing the density of each chemical species (the ϕ parameter in the SI). The density of head and tail densities are displayed using Paraview (a standard visualization software).

8. In line 247: “The barrier decrease for this pathway is significant.”—how is significance determined in this case?

“Significance” here refers to the magnitude (as opposed to statistical significance) of the reaction rate. Comparing the barriers of the rate-limiting steps, the difference is approximately 5κ , with $\kappa \sim 20kT$. This corresponds to an increase in the reaction rate by a factor $\sim \exp(100)$, which is very large, i.e., we expect a large difference in the fission rate. We have modified the quoted statement to make it clear that we mean that the change is large (more than several kT).

9. Suggestion for improvement: In line 278: “However, ultrastructural observations of mitochondrial fission necks suggest that transient contacts between constricted mitochondrial membranes are possible due to their close proximity^{16–18,64}.” Some of these referenced studies show that mitochondrial membranes constrict simultaneously, even in the most constricted states, which contradicts the interpretation/statement above. Could the authors comment on this further?

We agree that the statement is ambiguous. What we meant to say was that the distance between mitochondrial membrane during the constriction becomes small enough to enable spontaneous (yet probabilistic) transmembrane contact formation. We rewrote accordingly.

When a double membrane system is constricted from the outside, both membranes are expected to constrict simultaneously, though maintaining a close distance due to hydration repulsion between them. Therefore, one would expect that the inner membrane would accumulate more curvature stress. The question we are addressing in our work is how this stress is released, as there are two possible outcomes: the canonical sequential fission path, through hemifission of the inner tube, or a transient contact between membranes. Our calculations indicate that the transient contact is preferable when constriction proceeds at slow rate^b or remains far from extreme constriction conditions, such as in the case of mitochondrial fission. Such transient contacts are metastable structures difficult detectable in cells.

We have modified the text to clarify that this is expected for simultaneously constricting membranes.

Reviewer #2

1. Rapid changes in osmolarity, resulting from water flow, should induce membrane bulging as water moves inside a closed membrane system. An important question to address is whether this membrane expansion contributes to the fission process. Authors should elaborate on this point to provide a more comprehensive understanding of the phenomenon.

^b If the constriction is slow, the system spends significant time at small constriction and can form a transmembrane contact before the alternate classical mechanism at higher constriction can be pursued.

Rapid changes in osmolarity do not necessarily induce fluxes in the enclosed volume, it would depend on the geometry, surface/volume ratio, and the speed of the osmoticant application. Please note that uniform constriction of a uniform cylinder can be easily achieved in our system (see Fig. 3b and Supplementary Fig. 2 and 3). Nevertheless, we believe that the periodic bulges, like those that we observe in tubes even without fission, are commonly brought about by tension gradients, such as that produced by the flow in our system¹³ and do not require the motion of water into the closed membrane.

2. In Fig. 1a, authors should elaborate more on how they differentiated between a single, double or multi-membrane system in the fluorescence experiments? Is it possible to mention the change in tube diameters or change in fluorescence intensities in the text? How the changes between DM and multi-membrane systems can be distinguished? In Fig. 1b legends, does n=3 represent the number of tubes or on the number of times experiment was performed?

The suspended nanotube system is not new to the field. We have added a reference to previous publications about this system where we specifically address the lamellarity analysis of the NTs. We have added the changes in tube diameters to the text. Prior to the experiment, the lamellarity of the tubes is assayed. While multilamellar NTs have a much lower probability to be formed in this system¹⁴ we choose the tubes according to previously established range of fluorescence intensities. “n” represents both, the number of experiments and tubes (1 tube per experiment). We have clarified these points in the new version of the text and added a Supplementary Fig. 1b showing the lamellarity analysis.

3. It's not clear how the authors distinguish between inner membrane fission versus outer membrane fission based on fluorescence microscopy. This should be described in more detail. Also, the result that the inner membrane undergoes fission before the outer membrane should be supported by cryoEM images showing an intermediate state with the inner membrane separated and the outer membrane intact.

The referee is correct that it is impossible to distinguish outer versus inner membrane fission by fluorescence microscopy. This statement is now corrected in the new version of the text to avoid ambiguity. However, we have never observed a structure resembling outer membrane fission by cryoEM, while inner membrane structures inside a tube are a common phenotype in “dynamic snapshots” experiments. Indeed, our Supplementary Fig. 2c shows how an inner membrane tube retracts inside an outer tube. Besides, our calculations of the energy barriers for different pathways indicate that outer tube fission is energetically unfavorable. The only plausible mechanism that we may envision for such an outcome is poration of the outer membrane (hemifission would not be possible due to the presence of the inner tube). As explained above (reviewer 1, point 2), membrane poration would require a very high membrane tension/constriction, incompatible with the radii we experimentally observe by cryoEM and fluorescence microscopy. Taken together all these arguments favor fission of the inner tube as the first event in double membrane fission process. We have clarified this point in the text.

4. In Fig. 2, can the measurements on outer membrane diameters be provided? Does it remain the same or change during constriction? A comparison of outer diameter of SM and DM would be beneficial.

The outer membrane constricts. The outer radius is set by (i) its tendency towards the equilibrium radius (controlled by the tension and bending energy) and (ii) hydrophobic repulsion against the inner tube. As the inner tube constricts, the repulsion decreases, and the outer tube is allowed to constrict closer to its preferred radius.

We have added the measurements of the radii, as well as a short discussion, summarizing the above to the new version of the text (Fig. 2, caption).

5. Estimating the exact dimensions of nanotubes requires a single supported lipid bilayer (SLB) [1]. However, this can be challenging to compute when authors employ microsphere beads to create nanotubes, as these microbeads can introduce excess membrane, complicating the analysis. Therefore, the authors should consider determining tube dimensions during water-induced membrane fission using cryo-electron microscopy (cryoEM).

We note that the SLB calibration, as described in Materials and Methods section (see also^{8,14,15}) was made in a separate experiment. It is used to correlate the fluorescence signal from the membrane to its area. The reservoir on the bead is not related to that calibration. The method yielded correct radii for Drp1-driven and Dynamin 1 driven constriction^{8,14}. Hence, we are confident that our radii measurements are correct.

6. Starting tube diameter also influences membrane fission based on the protein-mediated fission literature. In the double membrane system, the inner tube would have higher curvature stress and so may undergo fission preferentially. Are the tube diameters for the inner double membrane tube and single membrane tube (figure 1a) similar? These should be the same to conclude that inner double membrane tube undergoes preferential fission. The authors should consider analyzing the dependence on tube diameter to support their conclusion that the inner membrane undergoes preferential fission.

Please also see our response to the comment number 3, where we demonstrate that fission-by-fusion is preferred to single-tube fission or the classical, sequential mechanism in double-tube fission (at fixed inner-tube radius or fixed tension). As we already pointed out, our experimental data reveal that fission of the inner tube requires less energy (for it undergoes fission at lower curvature) than the single membrane when constriction happened at slow rate (i.e. in osmotically induced fission in Fig. 1). This discovery was key, providing the overall study hypothesis. All the dataset, from osmotically induced fission in Fig. 1, fission induced by flow in Fig. 3, to the SCFT calculations support our claim of inner membrane fission undergoing fission through the fission-by-fusion mechanism. We already provided the radii analysis that support our claim in Fig. 1.

7. *Line 179 Panel 2c is missing from Figure 2.*

Thank you for pointing out this typo.

8. *The current result on the formation of worm-like micelle under high tension seems qualitative as tension cannot be measured in the current setup. Is there a way to quantitate the threshold tension over which WLM is readily observed? A quantification of the appearance WLM structure in cryo-EM data would help. The authors should compare tubes formed with bead rolling versus tubes formed with blotting-induced fluid flow by cryoEM if possible.*

Local membrane tension for a cylindrical geometry can be predicted from the tube radii¹⁶. We have now added this analysis as an inset in Supplementary Fig. 1e. The local tension threshold for WLM formation is in good agreement with theoretical predictions, further confirming the validity of the SCFT approach to this kind of phenomena.

9. *Can the system be tested with the addition of proteins that cause membrane fission like dynamin? It would be interesting to know how the system behaves in more physiologically relevant conditions.*

It definitely can and is in progress, but it is a separate project, which clearly goes beyond the scope of this manuscript. We note that protein addition to the system would not necessarily make the manuscript more physiologically relevant. The present manuscript is focused on fundamentals of the lipid behavior, relevant for **all** protein machineries, while each protein brings its own specifics (due to its geometry, mechanics etc.) to the fission process. This approach allows us to predict the different factors that may affect membrane rearrangements in the cell, which are more than the presence of the external constriction protein. For example, our model predicts the effect of increase in local tension as an important regulatory factor in the process. Transient contacts between membranes may also lead to fast lipid exchange between the membranes, which will affect the mechanical parameters of both membranes. Overall, we believe that our result will make a solid foundation for future studies of the double membrane fission process, that has not been addressed to date.

Reviewer #3

1. *As claimed by the authors on line 85 and 86, the SCFT are mainly applied to small scale membrane transformations in the past, but the scale of magnitude of their simulation system is much larger. As a result, is the method still sound and rigorous in the current system? What is the limitation of the SCFT method? And what are the assumptions of this theoretical treatment? To validate the theoretical method, one typically needs to use the precise experimental measurements as a benchmark. In this regard, what is the experimental benchmark in the current system? And can the authors provide any error estimation of their theoretical method?*

Scale: Previous SCFT membrane calculations have, indeed, focused on smaller systems. The only difference between applying SCFT to small vs. large systems is the computational resources required. We were able to study such large systems, without compromising spatial or contour (along the molecule) resolution by using one of the largest supercomputers in Europe, JUWELS at the Jülich Supercomputing Centre, to employ more computational resources.

Assumptions/limitations: SCFT (as with any method) has limitations, but we do not expect these to lead us to qualitative or large quantitative discrepancies. We have addressed the limitations of SCFT in the SI (where SCFT is discussed) and added a discussion of the limitations detailed below to the discussion section. Our responses are as follows:

(i) SCFT is a mean-field theory and lacks random composition fluctuations. These have been found only to be significant for disordered or weakly segregated systems, but as lipid tails segregate quite strongly from heads/water, this is not a significant concern.

(ii) We treat lipids as Gaussian chains, which ignores their stiffness. We expect the general behavior to be unaffected, due to the universality of self-assembly in amphiphilic polymers, however, there may be quantitative differences. These differences are partially accounted for by mapping our predictions onto real lipid behavior by scaling our free energies by the bending energy.

Validation with experiments. There are qualitative and quantitative predictions, both types are of great value for our understanding of biological phenomena. In this case, the modelling yields clear qualitative predictions on the topology (topological classes) of double-membrane fission. Pairing with experiments was constructed along this line of predictions. We note that this is the first-in-class comprehensive study of the pathways of double-membrane fission. Determination of topological classes is a reasonable start, and it already reveals the intriguing complexity, with clear links to biology. Hence, we believe that the modeling and experiment presented here do constitute a stand-alone manuscript, while more quantitative analyses shall definitely follow.

2. The authors claimed that the HF pathway breaks the canonical pathway into several intermediate states. This way, the transition between the intermediate states requires less energy, lowering the energy barrier of the rate-limiting steps and as well as the overall energy. Does this energy reduction come from entropy or enthalpy? In Fig3 a, the energy at $\alpha=0$ (fission not started) is lower than $\alpha=1$ (fission completed). Is this attributed to the additional membrane-water interface created by the fission? To provide a clear physical picture of this noncanonical pathway of membrane fission, the authors need to specify the different sources of energy change corresponding to different stages of the fission.

Generally, breaking up a pathway into multiple metastable intermediates will increase the rate, which is dictated by the largest free-energy barrier along the path.

To a first approximation, the difference in free energy between the states at $\alpha=0$ and 1 mainly comes from the fact that the membranes are under tension and have pulled apart, thus creating less membrane ([smaller area] \times [positive free energy per unit area i.e. tension]).

The main source of the decrease in free energy is the hydrophilic-hydrophobic repulsion in our coarse-grained model, as the head-tail interface contributes greatly to the membrane tension.

The tension and bending energies and other contributions to the free energy arise from a complex interplay of lipid entropy (number of conformations of the head and tail, as well as translational entropy) and energy (number of head/water-tail contacts that varies with area and curvature). All of these contributions are captured by our model, described in the SI.

3. For the hemi-fusion pathway, the pore formation would expose the hydrophobic tails of the lipid, which is energetically unfavorable in general. How was this energy penalty compensated in the model? One possibility is that the formation of the pore promotes the flip-flop of the lipids, which is not depicted in the model. This leads to a more fundamental question of the lipid exchange during the fission pathway. The authors need to either incorporate this effect in their theoretical model or prove that the flip-flop/lipid exchanges do not notably contribute to the fission of double membrane.

When a pore forms, the lipids rearrange to line the rim with head groups, preventing the tail groups from being exposed and minimizing the line tension of the pore's rim. Rearrangements such as this and flip-flops are "automatically" included in SCFT, as the calculation of lipid statistics imposes lipid equilibrium, given the concentration profiles. There is no equilibration time. We have modified the discussion of SCFT in the SI, to clarify this.

4. In a broader scope, the main conclusion of this work stems from theoretical predictions. The problem is: Both the theory and the calculation in the work are built upon several assumptions of the membrane shape and dynamics, which are not first-principle. To substantiate their conclusion, the authors need to experimentally validate either the assumption or the prediction of the theoretical model, or both. As it currently stands, it is difficult to appreciate exactly how experiments validate the essence of the model. For instance, what is the statistics of cryoEM data in Fig 3c-d? It is just one particular image, or a general phenomenon. If the latter, then what is the sample size, and what is the percentage of the cases that clearly shows the fission/the inner-outer membrane contact? What is the proper experimental control for Fig. 3c-d? How can we be sure that the cryoEM experimental procedure does not introduce any perturbation on the double membrane shape? Without the rigorous experimental validations, the conclusion can only be viewed as a hypothesis, which greatly decreases the significance of the work.

First, we would like to respectfully point out that our approach to studying double-membrane fission is based upon decades of similar studies of SM fission, where the same combination of experiments (structural analyses based upon cryo-EM and fluorescence-microscopy combination^{17,18}) and theoretical and computer modeling¹⁹⁻²⁸ have been successfully used. Those, in turn, are intimately linked to membrane fusion field. Thus, we see no need for an in-depth justification of the methods used, for they all have been verified in single-membrane systems. But the referee is correct that the description of the datasets was not complete and the cryoEM dataset could be expanded. Following this advice, we have now amplified the cryoEM image gallery presented in new Supplementary Fig. 2 and 3 to further support our claims. As to the statistics on the sample, please note that this is a highly stochastic process based on fast extension of the reservoir membrane. We have no means to control the reservoir extension or the force applied to extend the nanotubes, therefore a statistical analysis of the sample has no physical interpretation and should remain qualitative. However, each time we observe the local thinning of the NTs we

also detect membrane scission and WLM formation. We have now added a qualitative analysis of membrane local tension threshold for WLM formation (Supplementary Fig. 1e, inset), which show high correlation between predicted values and the experimental ones.

On the other hand, the theoretical model is based on assumptions about the statistical behavior of lipids. Although we use a coarse-grained, simplified representation of a lipid (see also above) such coarse-grained models have proven successful in predicting a multitude of aspects amphiphilic self-assembly in synthetic and biological systems^{19–21} including pore formation^{22,23} and fusion^{19,24–28}. In the last 20 years of research, comparison between coarse-grained models and experiments as well as between different levels of coarse-grained models have established their scope and limitations. The lipidic aspects of change of membrane topology such as pore formation, fusion, and fission belong to universal aspects where the use of coarse-grained models is well established. Thus, we are confident that our predictions are valid because of the numerous examples cited where SCFT has been successfully used to predict lipid behavior. This confidence is further bolstered by our own calculations finding the canonical pathway, which shows that it can describe large, complicated rearrangements that are known to take place.

References

1. Chernomordik, L. V. *et al.* The shape of lipid molecules and monolayer membrane fusion. *Biochim. Biophys. Acta (BBA) - Biomembr.* **812**, 643–655 (1985).
2. Kozlov, M. M., Leikin, S. L., Chernomordik, L. V., Markin, V. S. & Chizmadzhev, Y. A. Stalk mechanism of vesicle fusion. *Eur. Biophys. J.* **17**, 121–129 (1989).
3. Zimmerberg, J. & Chernomordik, L. V. Membrane fusion. *Adv. Drug Deliv. Rev.* **38**, 197–205 (1999).
4. Chizmadzhev, Y. A., Cohen, F. S., Shcherbakov, A. & Zimmerberg, J. Membrane mechanics can account for fusion pore dilation in stages. *Biophys. J.* **69**, 2489–2500 (1995).
5. Chernomordik, L. V. & Kozlov, M. M. PROTEIN-LIPID INTERPLAY IN FUSION AND FISSION OF BIOLOGICAL MEMBRANES *. *Annu. Rev. Biochem.* **72**, 175–207 (2003).
6. Chernomordik, L. V. & Kozlov, M. M. Mechanics of membrane fusion. *Nat. Struct. Mol. Biol.* **15**, 675–683 (2008).
7. Harmandaris, V. A. & Deserno, M. A novel method for measuring the bending rigidity of model lipid membranes by simulating tethers. *J Chem Phys* **125**, 204905 (2006).
8. Mahajan, M. *et al.* NMR identification of a conserved Drp1 cardiolipin-binding motif essential for stress-induced mitochondrial fission. *Proc. Natl. Acad. Sci.* **118**, e2023079118 (2021).

9. Reichert, A. S. & Neupert, W. Contact sites between the outer and inner membrane of mitochondria—role in protein transport. *Biochim. Biophys. Acta (BBA) - Mol. Cell Res.* **1592**, 41–49 (2002).
10. Friedman, J. R. Mitochondria from the Outside in: The Relationship Between Inter-Organellar Crosstalk and Mitochondrial Internal Organization. *Contact* **5**, 25152564221133268 (2022).
11. Pérez-Jover, I. *et al.* Allosteric control of dynamin-related protein 1-catalyzed mitochondrial fission through a conserved disordered C-terminal Short Linear Motif. *Res. Sq.* rs.3.rs-3161608 (2023) doi:10.21203/rs.3.rs-3161608/v1.
12. Cheng, Y., Grigorieff, N., Penczek, P. A. & Walz, T. A Primer to Single-Particle Cryo-Electron Microscopy. *Cell* **161**, 438–449 (2015).
13. Lyu, J. *et al.* Dynamics of pearling instability in polymersomes: The role of shear membrane viscosity and spontaneous curvature. *Phys. Fluids* **33**, 122016 (2021).
14. Galvez, J. M. M., Garcia-Hernando, M., Benito-Lopez, F., Basabe-Desmonts, L. & Shnyrova, A. V. Microfluidic chip with pillar arrays for controlled production and observation of lipid membrane nanotubes. *Lab a Chip* **20**, 2748–2755 (2020).
15. Espadas, J. *et al.* Dynamic constriction and fission of endoplasmic reticulum membranes by reticulon. *Nat Commun* **10**, 5327 (2019).
16. Roux, A. The physics of membrane tubes: soft templates for studying cellular membranes. *Soft Matter* **9**, 6726–6736 (2013).
17. Antonny, B. *et al.* Membrane fission by dynamin: what we know and what we need to know. *The EMBO Journal* **35**, 2270–2284 (2016).
18. Frolov, V. A., Escalada, A., Akimov, S. A. & Shnyrova, A. V. Geometry of membrane fission. *Chem Phys Lipids* **185**, 129–140 (2015).
19. Müller, M., Katsov, K. & Schick, M. Biological and synthetic membranes: What can be learned from a coarse-grained description? *Physics Reports* **434**, 113–176 (2006).
20. Zhang, P.-W. & Shi, A.-C. Application of self-consistent field theory to self-assembled bilayer membranes. *Chinese Phys B* **24**, 128707 (2015).
21. Ting, C. L. & Müller, M. Membrane stress profiles from self-consistent field theory. *The Journal of Chemical Physics* **146**, 104901 (2017).
22. Ting, C. L., Appelö, D. & Wang, Z.-G. Minimum Energy Path to Membrane Pore Formation and Rupture. *Physical Review Letters* **106**, 309 (2011).

23. Ting, C. L., Awasthi, N., Müller, M. & Hub, J. S. Metastable Porepores in Tension-Free Lipid Bilayers. *Physical Review Letters* **120**, 128103 (2018).
24. Ryham, R. J., Klotz, T. S., Yao, L. & Cohen, F. S. Calculating Transition Energy Barriers and Characterizing Activation States for Steps of Fusion. *Biophys J* **110**, 1110–1124 (2016).
25. Daoulas, K. C. & Müller, M. Exploring thermodynamic stability of the stalk fusion-intermediate with three-dimensional self-consistent field theory calculations. *Soft Matter* **9**, 4097–4102 (2013).
26. Müller, M. & Schick, M. An Alternate Path for Fusion and its Exploration by Field-Theoretic Means. in vol. 68 295–323 (Current Topics in Membranes, 2011).
27. Katsov, K., Müller, M. & Schick, M. Field Theoretic Study of Bilayer Membrane Fusion. I. Hemifusion Mechanism. *Biophysical Journal* **87**, 3277–3290 (2004).
28. Katsov, K., Müller, M. & Schick, M. Field Theoretic Study of Bilayer Membrane Fusion: II. Mechanism of a Stalk-Hole Complex. *Biophysical Journal* **90**, 915–926 (2006).

REVIEWER COMMENTS

Reviewer #1 (Remarks to the Author):

I am generally pleased with the revisions made to the manuscript. The additional electron microscopy (EM) images and the inclusion of further explanations throughout the document contribute to a clearer understanding. The additional text clarifications sprinkled throughout the manuscript help place these results within the broader context of double membrane organelle fission.

Reviewer #2 (Remarks to the Author):

The authors have addressed all my concerns in the current revised manuscript. Therefore, I accept this updated manuscript.
Congratulations!

Reviewer #3 (Remarks to the Author):

The authors sufficiently addressed the major points on theoretical approaches. I have no further issues in that regard. My main concern lies in the experimental validation. As I understood the exp data in Fig 3 correctly, there are only 10 data points from which the authors determined the mean and SD. First, the sample size of 10 data points is too small for statistical analysis; I am not sure how meaningful the calculated mean and SD are. Second, it is not clear from the experiments how often the inner membrane forms the contact with outer membrane en route to membrane fission. In other words, is it the case that these inner-outer membrane contacts emerged on every double-membrane tubule every time the experiment was performed? If so, then how many sets of the independent experiments were performed? If not, then what is the fraction of the “successful” experiment? Is it 1 out 10 membrane tubules, or 5 out 10 etc? Without this information, it is uncertain how to judge the validity of experimental claims and its connection to the prediction from self-consistent field theory.

Point-to-point response to reviewer 3 comments

We thank the reviewers for their feedback on our manuscript. We next respond to the comments from reviewer 3:

My main concern lies in the experimental validation. As I understood the exp data in Fig 3 correctly, there are only 10 data points from which the authors determined the mean and SD. First, the sample size of 10 data points is too small for statistical analysis; I am not sure how meaningful the calculated mean and SD are.

SD is defined as the deviation of the data from the mean and its meaning for small sets of data is well defined (in statistics as well as natural sciences, see e.g. *Simplified Statistics for Small Numbers of Observations*, R. B. Dean and W.J Dixon, *Anal. Chem.* 23, 636 (1951)). The reviewer could potentially be confused about the specific meaning of the data in Figure 3. We intended to use descriptive statistics to characterize the micrographs seen in Figure 3. Mainly, we present the mean of 10 random measurements made in the regions pointed by arrows.

However, we appreciate the reviewer's inquiry about the statistics of our measurement. Therefore, we now include the mean and average of 111 measurements of SM thickness made over 18 tubes in 3 independent experiments. Please note that this number gives the same mean and similar SD values for the single membrane tubes (measured as peak-to-peak distance of the plot profile normal to the membrane, as shown in Supplementary Figure. 1d): 5,1 +/- 0,5, mean+/-SD, confidence interval 0,09 at 95% confidence level. We hope these numbers convince the reviewer about the validity of the distance measurements in Figure 3.

Second, it is not clear from the experiments how often the inner membrane forms the contact with outer membrane en route to membrane fission. In other words, is it the case that these inner-outer membrane contacts emerged on every double-membrane tubule every time the experiment was performed? If so, then how many sets of the independent experiments were performed? If not, then what is the fraction of the "successful" experiment? Is it 1 out 10 membrane tubules, or 5 out 10 etc? Without this information, it is uncertain how to judge the validity of experimental claims and its connection to the prediction from self-consistent field theory.

We would like to note that the cryoEM observations of transient intermediates are challenging both, experimentally and in their quantitative interpretation, as we aim at the extremely short-lived intermediates of the dynamic process. The fact that we do detect intermembrane contacts at all is already indicative of the existence of such structures and the complex interplay between bilayers of the multilayered system, as further explored by the modeling. As indicated in the manuscript, the formation of SM tubes prevails in our experimental system. Out of 110 tubes detected in 7 independent experiments, 14.5% were DM tubes. We note that the tubes were produced by flux, hence we look at snapshots of tubular structures under constriction force, driving the tubes towards fission. The theory predicts that the DM tubes undergo fission earlier (at smaller constriction), which might explain the prevalence of SM tubes as we suggested in the manuscript. It also follows that capturing metastable DM configurations at close-to-fission curvatures is extremely difficult. Yet, we obtained 16 of such tubes.

In 3 out of 16 DM tubes we did not detect any intermembrane interaction. For the rest of the tubes, we compared the half of the distance between the outmost lipid head regions of the DM

with the mean value of the SM thickness. In 4 DM tubes, we detected regions of closely apposed bilayers with intermembrane space ranging from 1.7 to 0.6 nm (with equal or not equal variances assumed at 0.05 level). In the remaining 9 tubes, we detected intermembrane contacts that we define as localized membrane regions with intermembrane distance < 0.5 nm. Interestingly, in 6 out of 9 tubes, the DM thickness was smaller than the sum of 2 SM thicknesses (mean $DM/2 - \text{mean SM} < 0$, with equal or not equal variances assumed at 0.05 level), indicative of the formation of a HF intermediate. This data is now included in the manuscript and also disclosed in the Source Data file attached.

We would like to point that the major prediction from self-consistent field theory is that at relatively low membrane tension intermembrane contacts will form before the critical constriction of the inner tube is reached (corresponding to the inner membrane radius $< 2 \text{ nm}^{1,2}$). This is the case in all such tubes detected in our experiments. Therefore, our experimental data fully support the theoretical predictions.

1. Kozlovsky, Y. & Kozlov, M. M. Membrane Fission: Model for Intermediate Structures. *Biophys J* **85**, 85–96 (2003).
2. Frolov, V. A., Escalada, A. & Akimov Sergey A. and Shnyrova, A. V. Geometry of membrane fission. *Chem Phys Lipids* **185**, 129–140 (2015).

REVIEWERS' COMMENTS

Reviewer #3 (Remarks to the Author):

Now the authors incorporated the rigorous statistics of the experimental data. With that, I think the paper is ready for publication.